# Pru p 9, a new allergen eliciting respiratory symptoms in subjects sensitized to peach tree pollen

Miguel Blanca[1], Laura Victorio Puche[2], María Garrido-Arandia[3], Laura Martin-Pedraza[4], Alejandro Romero Sahagún[3], José Damian López-Sánchez[5], Carmen Galán[6], Antonio Marin[7], Mayte Villaba[4], Araceli Díaz-Perales[3], Maria Luisa Somoza[1]*

1 Allergy Service, Hospital Universitario Infanta Leonor, Madrid, Spain, 2 Allergy Service, Hospital General Universitario Morales Meseguer, Murcia, Spain, 3 Centro de Biotecnología y Genómica de Plantas (UPM-INIA), Madrid, Spain, 4 Departamento de Bioquímica y Biología Molecular, Facultad de Químicas, Universidad Complutense de Madrid, Madrid, Spain, 5 Allergy Service, Hospital Clínico Universitario Virgen de la Arrixaca, Murcia, Spain, 6 Departamento de Botánica, Universidad de Córdoba, Córdoba, Spain, 7 Centro de Salud de Blanca, Blanca, Murcia

* mlsomoza@yahoo.com

**Data Availability Statement:** All relevant data are within the paper and its Supporting Information files.

## Abstract

Peach tree (PT) pollen sensitization is highly prevalent in subjects living in areas where this tree is widely cultivated. None of the allergens responsible for these sensitizations have been identified so far. Our aim was to identify the most relevant PT pollen allergens and analyze their capacity for inducing respiratory symptoms. We studied sixty-two individuals sensitized to PT pollen who developed symptoms after its exposure. The IgE binding profile on peach pollen extract by means of immunoblotting using sera from these subjects was analyzed. Protein extract was fractionated by anion-exchange chromatography and HPLC, fractions run in SDS-PAGE and proteins were identified from IgE-binding bands by mass spectrometry. Several allergenic proteins in the PT pollen extract were recognized by patients' IgE: a glucan endo-1,3-beta-glucosidase-like, a polygalacturonase, an UTP-glucose-1-phosphate uridylyltransferase and a PR-1a protein. This PR-1a protein is a novel allergen frequently recognized with a molecular mass of 18 kDa, named as Pru p 9 following the WHO-IUIS nomenclature. Skin Prick Test (SPT) performed with this allergen was positive in 41% of the PT pollen-sensitized clinical cases. Most of them had rhinitis or rhinoconjunctivitis, but a significant percentage experienced asthma with seasonal symptoms during the period of PT flowering. Nasal Provocation test (NPT) with Pru p 9 was positive in all cases with positive SPT to this new allergen eliciting nasal symptoms similar to those challenged with PT pollen. We demonstrate that PT pollen can induce sensitization and allergy in an exposed population, being Pru p 9 a relevant allergen responsible of respiratory symptoms. Considering the extensive peach worldwide production with a large number of people involved, our results add a great value for the diagnosis and management of subjects allergic to this pollen.

**Funding:** This work was supported by the grants PI17/00615 and SAF2017-86483-R from the Ministerio de Economía y Competitividad and ISCIII co-founded by Fondo Europeo de Desarrollo Regional – FEDER for the Thematic Networks and Co-operative Research Centres: ARADyAL (RD16/0006/0014, RD16/0006/003 and RD16/0006/0024). And also from Allerscreening Project (H2020-NMBP-X-KET-2017. 768641 - AllerScreening). The proteomic analysis was performed in the Proteomics Service of Complutense University of Madrid, a member of ProteoRed and is supported by grant PT17/0019, of the PE I+D+i 2013-2016, funded by ISCIII and ERDF. The funders had no role in study design, data collection and analysis, decision to publish, or preparation of the manuscript.

**Competing interests:** The authors have declared that no competing interests exist.

## Introduction

Pollen grains are amongst the biological agents most frequently involved in allergic diseases all over the world [1], and can be anemophilous or entomophilous [2]. Many plant pollens from fruit trees are entomophilous and therefore considered of low allergenic interest, except for subjects occupationally exposed [3, 4]. Cases of allergic reactions to pollen from fruit trees and ornamental plants have been reported, although mainly limited to workers directly exposed or pollen that is transported by airborne particles [5–9]. However, in certain regions of the world fruit tree pollens can be relevant not only in exposed individuals but also in the general population, as occurs with palm tree pollen [10, 11]. The report of symptoms in a high number of PT pollen-sensitized people, including children after being exposed to the clothes of their parents who have been working in the crop fields of fruit trees led us to undertake this study [12].

In certain cases, pollens from cultivated and non-cultivated plants can be relevant for subjects exposed at different agricultural sites [4, 5]. In the case of cultivars of fruit trees, although entomophilous, pollen grains can be transported by the wind for short distances [9, 13]. This transport can be facilitated by the presence of pollutants or the disruption of pollen in smaller submicronic particles [2]. In the case of the PT, allergens can also be released from other plant organs, such as leaves or stems [14, 15]. Allergy to fruit and vegetal pollens has been reported with grapes, chestnut, strawberries, paprika, tomato, oranges, satsumas and apples, among others [5, 7, 9, 16, and 17]. Occupational asthma has been reported for the *Rosaceae* family [18, 19], and for other pollens to which people can be sensitized [8, 16]. Our purpose was to study a group of subjects sensitized to PT pollen to determine which allergens were involved, with further characterization and confirmation of the clinical relevance by NPT. A novel relevant and specific allergen from PT pollen responsible of symptoms in a group of sensitized subjects directly or indirectly exposed to PT pollen was identify as a PR-1a protein and included into *WHO/IUIS Allergen Nomenclature* Database as Pru p 9 [20,21].

## Materials and methods

### Cases' selection

Between January-June 2018, we evaluated 310 consecutive subjects who were older than 15 y. o. and lived in an area of PT cultivars. The allergological study included a detailed questionnaire plus SPT. Cases were divided in two groups: A) SPT positive to PT pollen and Pru p 9 and B) SPT positive to PT pollen but negative to Pru p 9. A control group of pollen sensitized subjects from a region where there are no PT cultivars was included in order to verify the capacity of PT pollen for inducing symptoms.

All the participants had a personal interview. The questionnaire included date and place of birth, the time living in the region, occupation, and history of allergic diseases like rhinitis, conjunctivitis, asthma, atopic dermatitis and urticaria. Those reporting symptoms were also questioned about intensity, duration, relation with occupational exposure and the month(s) of the year the symptoms occurred. The clinical evaluation was made independently by two physicians. After completing a first interview that included skin testing, the subjects were re-evaluated again to make the records more precise. Prior to these individual evaluations, various educational sessions were given to village citizens, school teachers and health care professionals, as well as to the individuals participating in the study.

A written informed consent was signed by the participants or their parents prior to the inclusion. The study was approved by our institutional Ethics Committee: *Comité de ética de la investigación. Hospital General Universitario Gregorio Marañón.*

## Skin Prick Test and subjects' sera

Skin Prick tests [22, 23] were performed with commercially available drug products (ALK-Abelló®, S. A., Madrid, Spain) of the species *Phleum pratense*, *Cupressus arizonica*, *Olea europaea*, *Platanus acerifolia*, *Parietaria judaica*, *Artemisia vulgaris*, *Plantago lanceolata*, *Chenopodium album* and *Salsola kali*. The drug product was prepared, following the Good Manufacturing Practice requirements, by extracting the corresponding pollen with phosphate buffer at a 10% w/v ratio. After diafiltration (5kD cutoff), the extract was formulated in 50% glycerol and 0.51% phenol and adjusted to the potency indicated in the corresponding labelling [24].

For sera obtaining, 10 ml of blood was drawn by venipuncture from all the study participants in order to obtain sera and perform the immunological determinations mentioned below. After processing, the sera were kept at -20˚C until further use for allergen characterization. Two pool of sera were made: pool 1 (sera from 6 cases sensitized to PT pollen) and pool 2 (sera from 6 cases sensitized to other pollens but negative to PT pollen).

## Peach tree pollen protein extract and Pru p 9 purification

PT pollen (Iberpolen®, Spain) was defatted with dimethyl ether (1:5 [w/v]; 1 h; 4˚C) and, after drying, proteins were extracted with phosphate-buffered saline (PBS, 0.1M sodium phosphate, pH 7.0 and 0.5M NaCl; 1:5 (w/v), 1 h, 4˚C). After centrifugation (5000 rpm; 15 min; 4˚C), the supernatant was dialyzed against distilled $H_2O$ (cut-off point 3.5 kDa), and freeze-dried.

Protein extract was fractionated by anion-exchange chromatography on a Waters AccellTM Plus QMA Sep-PakR cartridge (pore size: 300 amstrong, Waters Corp, Milford, MA, USA). Elution was carried out with 20 mM ethanolamine, pH 9.0, and the retained material was then eluted with 0.75 mM NaCl in the same buffer (1 ml/min). In a second step, protein was purified by reversed-phase high-performance liquid chromatography (RP-HPLC) on a Nucleosil 300-C4 column (7x250 nm, particle size 5μm, Tecknokroma, Barcelona, Spain). The fraction was eluted with a linear gradient of acetonitrile in 0.1% trifluoroacetic acid (15% for 5 min, 15–85% for 120 min; 0.5 ml/min). Protein identification was made by UV detection at 280 nm.

Pru p 9 was purified from defatted mixture and extracted with PBS buffer (0.1 mM sodium phosphate pH 7.4, 1,5M NaCl; 1:5 (w/v); 1 h at 4˚C). Purified Pru p 9 was quantified by means of the commercial bicinchoninic acid (BCA) test (Pierce, Cheshire, UK) and purity was analyzed by SDS-PAGE.

## Mass spectrometry analysis

The protein samples were digested in trypsin solution. Briefly, proteins were reduced with 10mM DTT, alkylated with 55mM Iodacetamide and digested with recombinant Trypsin (Trypsin sequencing grade; Roche) overnight at 37˚C. After digestion, 1μl was spotted onto a MALDI target plate and allowed to air-dry at room temperature. Then, 0.6 μl of a 3 mg/ml of α-cyano-4-hydroxy-cinnamic acid matrix (Sigma) in 50% acetonitrile were added to the dried peptide digest and allowed again to air-dry at room temperature.

MALDI-TOF MS analyses were performed in a 4800 Plus Proteomics Analyzer MALDI-TOF/TOF mass spectrometer (Applied Biosystems, MDS Sciex, Toronto, Canada) at Proteomic Service from Complutense University of Madrid. The MALDI-TOF/TOF operated in positive reflector mode with an accelerating voltage of 20000 V. All mass spectra were calibrated internally using peptides from the auto digestion of trypsin.

The analysis by MALDI-TOF/TOF mass spectrometry produces peptide mass fingerprints and the peptides observed with a Signal to Noise greater than 12 can be collated and

represented as a list of monoisotopic molecular weights. Proteins ambiguously identified by peptide mass fingerprints, were subjected to MS/MS sequencing analyses using the 4800 Proteomics Analyzer (Applied Biosystems, Framingham, MA). So, from the MS spectra suitable precursors were selected for MS/MS analyses by Colision Induced Disociation (CID) using atmospheric gas and 1 Kv ion reflector operating mode. The precursor mass Windows isolation was +/- 4 Da. The plate model & default calibration were optimized for the MS-MS spectra processing.

For protein identification SwissProt DB 20170116 (553231 sequences; 197953409 residues) and NCBI DB with taxonomic restriction of "Viridiplantae" (1056156 sequences) were searched using MASCOT 2.3 (www.matrixscience.com) through the software Global Protein Server v 3.6 (ABSciex). Search parameters were: Enzyme: semiTrypsin; Carbamidomethyl Cystein as fixed modification and oxidized methionine as variable modification; Peptide mass tolerance: 50 ppm (PMF), 1 missed trypsin cleavage site; MSMS fragments tolerance 0.3 Da.

In all protein identification, the probability scores were greater than the score fixed by mascot as significant with a p-value minor than 0.05. The parameters for the combined search (Peak list from Peptide mass fingerprint and MS-MS spectra) were the same described above.

Proteins that were not identified by Mascot database searching were subsequently subjected to de-novo sequencing analyses, based on the fragmentation spectra of peptides, using DeNovo tool software (Applied Biosystems), tentative sequences were manually checked and validated. Homology search of the sequences was obtained by Blast (http://www.ncbi.nlm.nih.gov/BLAST).

## Analytical procedures: SDS-PAGE analysis

SDS-PAGE was carried out in 15% polyacrylamide gel, 0.1% SDS under reducing conditions with 2-mercaptoethanol (2-ME) as described [25]. Proteins were visualized with Coomassie blue staining. Molecular mass markers ranging between 75 and 15 kDa were used in one lane.

## Immunodetection assays with polyclonal antisera (pAbs)

To measure the IgE/IgG binding, PT pollen protein extract or purified proteins were separated by SDS-PAGE, and replica gels were electro-transferred onto polyvinylidene difluoride (PVDF) membranes. After blocking (Sigma–Aldrich, St. Louis, MO, USA), membranes were immunostained with allergic subject sera (1:5) or with specific polyclonal rabbit antisera against the most prevalent panallergens (profilin, thaumatin, nsLTP). After extensive washes, membranes were incubated with HRP-conjugated secondary antibody (Sigma, St. Louis, MO, USA) and signal developed with o-Phenylenediamine dihydrochloride (Sigma, St. Louis, MO, USA). The blotting inhibition assay was carried out in the same way, but the sera were previously incubated with the inhibitors (3 h at room temperature) before being added to the proteins transferred onto the membranes.

## IgE detection assays by immunoblotting with human sera

Immunodetection of proteins in nitrocellulose membranes was achieved as described by Pedraza et al (2016) by using a pool of sera with dilution (1:5). The binding of human IgE was detected with a mouse anti-human IgE monoclonal antibody (diluted 1:5000) kindly provided by ALK-Abelló® (Madrid, Spain), followed by HRP-labeled rabbit anti-mouse IgG (1:2500 diluted; Pierce, Rockford, Illinois). The signal was developed by the ECL-Western Blotting reagent, and detected in a luminescent imager analyzer LAS3000. Quantitation of the signal was performed in triplicate using the computer program Multigauge V3.0. Cases positive to at

least one of the prevalent pollens used in our panel but negative to PT pollen provided sera that were used in the immunoblotting assay.

### Nasal provocation test (NPT)

It was carried out as described elsewhere, with some modifications [26, 27]. Subjects in Group A and B were challenged in order to see the relevance of PT pollen and Pru p 9 in the induction of symptoms. The challenge was made in two sessions separated by 1 week: 1) on the first day subjects were challenged with PT pollen and on the eighth day they were challenged with Pru p 9. The allergen was applied spraying PT pollen extract on the head of the inferior turbinate and by instillation of the Pru p 9 solution on the inferior turbinate using a micropipette.

In order to verify the capacity of PT pollen and Pru p 9 for inducing symptoms we included a matched group of 6 subjects sensitized to *Phleum pratense*, *Olea europaea* and *Salsola kali* from a region where there are no PT cultivars (Madrid, Spain).

### Data analysis

For categorical variables we used the Chi square test. For quantitative variables we used the Student *t* test and for non-parametric quantitative variables we used the Mann-Whitney test. The association between PT pollen and the Pru p 9 wheal size was made by Pearson's correlation coefficient. These tests were done using the statistical Package SPSS 21.

## Results

### Clinical features of subjects

After evaluating 310 consecutive subjects 62 were SPT positive to PT pollen. This included individuals who had symptoms after working in PT cultivars or who, without working at the crop fields, have been directly exposed, and finally those who had symptoms during the PT pollination season but have been kept apart from the cultivars.

The mean age was 32 (+/-16) y.o. 61% were males. The mean number of pollens to which they were sensitized was 3(+/- 1.84). *Salsola kali* was the most frequently sensitizer (56%), followed by *Olea europaea* (49%), *Phleum pratense* (44%), *Cupressus arizonica* (40%), and *Parietaria judaica* (40%). According to symptoms, conjunctivitis appeared in the 47% of the cases, rhinitis in the 42% and asthma in the 14%.

### Cases sensitized to PT pollen specifically recognized a selective 18 kDa band of PT pollen extract

In an attempt to study the allergenic profile of subjects, proteins from defatted PT pollen were extracted in PBS. These were separated in SDS-PAGE and incubated with two different pools of sera: 6 sensitized to PT pollen (lane 1) and 6 sensitized to other pollens but negative to PT pollen (lane 2) (Fig 1). Sera from subjects sensitized to PT pollen (pool 1) recognized a selective IgE binding band of 18 kDa (M1). This band was not recognized by the subjects positive to pollens different to PT pollen (pool 2). In contrast, other high molecular mass bands (M2, M3, and M4) were also detectable in sera from pool 2. For identification of the pool 1 allergenic profile, the positive IgE bands were trimmed from SDS-PAGE and analyzed by peptide fingerprint-mass spectrometry (S1 Fig).

The M1 (18kDa) band rendered peptides corresponding to a Pathogenesis-Related (PR-1a) protein, which we have named as Pru p 9 following the guidelines of the allergen Nomenclature Committee (WHO-IUIS). Other PT pollen extract bands, M2 (25 kDa), matched to a

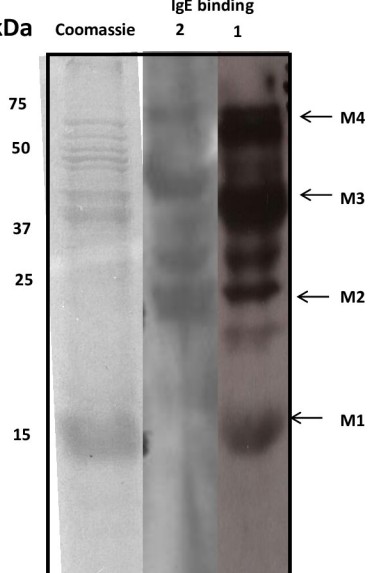

1: peach tree pollen allergic subjects
2: pollen allergic subjects.
M1: Pathogenesis related protein 1A.
M2: glucan endo-1,3-beta-glucosidase-like.
M3: Polygalacturonase.
M4: UTP-glucose-1-phosphate uridylyltransferase.

**Fig 1. SDS-PAGE and immunoblotting assay using a pool of sera of PT pollen sensitized patients (lane 1) and patients negative to PT pollen but sensitized to other pollens (lane 2).** 1: PT pollen allergic subjects. 2: other pollen allergic subjects. M1: Pathogenesis Related protein 1a. M2: glucan endo-1,3-beta-glucosidase-like. M3: Polygalacturonase. M4: UTP-glucose-1-phosphate uridylyltransferase.

glucan endo-1,3-beta-glucosidase; M3 (40 kDa), a polygalacturonase; and M4 (55 kDa), a UTP-glucose-1-phosphate uridylyltransferase. Data on amino acid sequence and classification groups are shown in Fig 2.

| Band | Sequence | Matched | Method |
|---|---|---|---|
| 1 | **MAFNTKLLLAICCVALVFTLVSANISK**EEIDGFVEEHNKARKEVGNKPLKWNTTLAQYAQEYADKR**VGDCAMEHSMGR**WGENLASG DGMSGAAATKYWVTEKEFYDEKSNKCVKDECGHYLAVVWGK**TTEVGCGISKCNNGQNYVVCSYDPMYQPEDERPY** | PR1-like protein | a, b |
| 2 | MATTKLLLSLFLLIQLAATAFAIGVNYGTLADNLPPPAQVANFLKTQTNIDKVKIFDANPDIIKAFANTNISLTITIPNGDIPSLTKLRAARRWV VDHVKPFYPQTKINYIAMGNEVLHWGDDNLKNSLVPAMR**TFHNALVR**EGIKDVK**VSTPHSLGIMLSSEPPSQGR**FRPEVIPLLTPMLQ FLRQTKGPFMVNPYPYFGWSPEKESFALFKPNK**GVHDQFTGK**SYTNMFDGLMDAVYSAAKAVGFGDVDLVAAETGWPSSCEFPVC SVQNAVDYNGHLIKHVESGK**GTPLMPNRK**FDTYIFALFNENQKPGPLAEKNWGLFKPDMTPVYNAGVMRNQQGGATPGPMVTTVA QPATPAAPAGPAKPAAPVKGGKKPKPATPAAPVAAGGGKKWCVAKPGATNQALQANIDYVCGKGIDCKPIQPGGTCFDNDVKAR**ASY LMNAYYQANGR**HDFNCDFSKTGQLTSADPSHGSCKYNA | glucan endo-1,3-beta-glucosidase-like | a |
| 3 | MGSKFIHGITWFLFLVAICAIKAKAISVDVVKFGAKGDGKTDDTK**AFTQAWTQACSER**QNNRYVIPKGTYIVGPVDFAGPCKAKTIHFKV DGTVQSSKKQSVTGGAHPNAWISFTQVNNLFISGDGIFDGQGFEGNCTKAKQCEQPPLNLIFAMVKDSHIQGITSNNSVGGHIGIYRSI NVTVDDVDIGIKGGEGILIEKSTNINIINTNIKILHDNCVTILDGNTGINIEKMTCSQGNGLGVSVLGNTGKEEPIKGVTVRNCTFSHTEGAI RIQSAAASNANIAISNLIFEDIIFDYLQNMAIILDQEHCPSKQCRTTNPSKVKVENVSFKNIKGTSVDPRIVILECGTAPDACKDIRFIDLR**V LVEGDDRLETQFR**CKNVKPAVAGHVDPAACNTRAVA | Polygalacturonase | a |
| 4 | MAAVATGNVDKLKSDVASLSQISENEK**NGFINLVSR**YVSGEEAQHVEWSKIQTPTDEVVVPYDGLAPTPEDPEEIKKLLDKLVVLKLNG GLGTTMGCTGPKSVIEVRNGLTFLDLIVIQIENLNNKYGSCVPLLLMNSFNTHDDTQKIVEKYSK**SNVQIHTFNQSQYPR**LVVEDFSPLP SKGQTGKDGWYPPGHGDVFPSLKNSGK**LDLLLSQGK**EYVFIANSDNLGAVVDLKILHHLIQKKNEYCMEVTPKTLADVK**GGTLISYE GR**VQLLEIAQVPDQHVNEFKSIEKFKIFNTNNLWVNLNAIK**RLVEADALK**MEIIPNPKEVDGVK**VLQLETAAGAAIRFFNHAIGINVPR**S RFLPVKATSDLLLVQSDLYTLQDGFVTRNSARKNPENPTIELGPEFKKVGSYLSRFKSIPSILELESLKVSGDVWFGAGVVLKGKVTITA KSGVKLEIPDNAVIANKDINGPEDL | UTP--glucose-1-phosphate uridylyltransferase | a |

**Fig 2. Identification of IgE binding bands.** IgE binding bands present in PT extract were identified by peptide mass fingerprint (method a) and MS/MS analysis (method b). Fragment matched by MS analysis are shown in red. The identification was carried out in the proteomics service of Complutense University of Madrid. (https://www.ucm.es/gyp/proteomica).

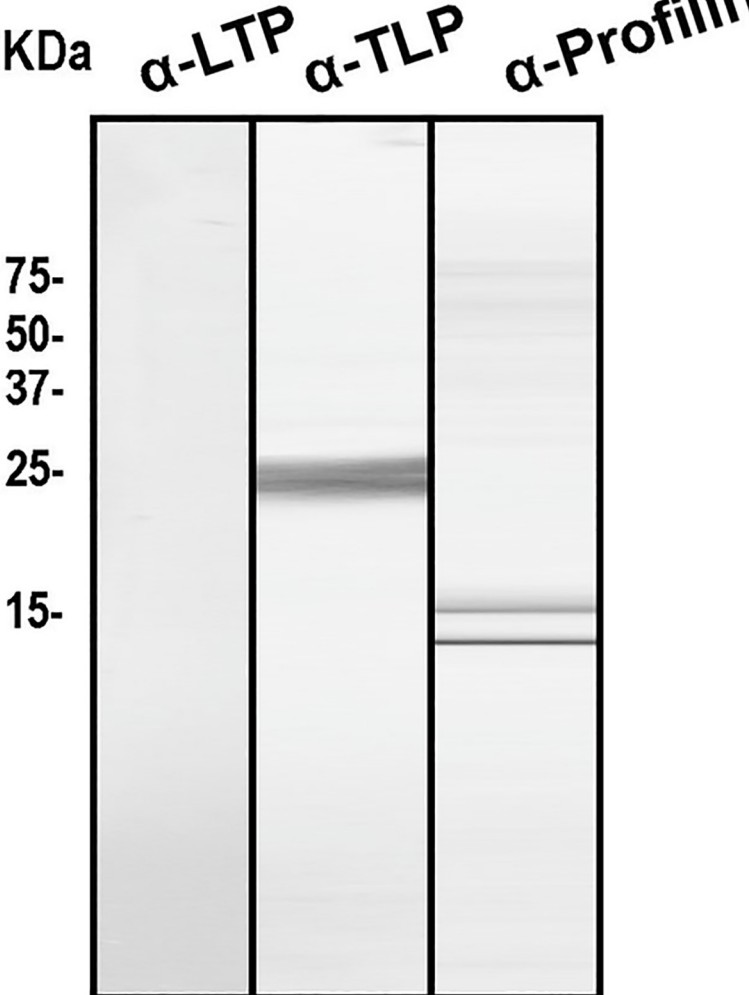

**Fig 3. SDS-PAGE and immunoblotting assay using rabbit polyclonal antisera to nsLTP, TLP and profilin.**

In order to complete the characterization of the PT pollen allergenic profile, the presence of the main panallergens was analyzed using a protein extract and immunodetected with different specific rabbit polyclonal antisera produced against the most relevant panallergens: anti-Pru p 3 (nsLTP), anti-Pho d 2 (Profilin) and anti-Pru p 2.2 (Thaumatin-like). As can be seen in Fig 3, IgG-reactive bands could be detected. PT pollen seems to contain profilin and thaumatin homologues, but no reactive bands corresponding to nsLTP was observed.

## Purification of Pru p 9

The 18 kDa IgE-reactive protein from PT pollen extract was fractioned by HPLC reverse phase column. The eluted protein at around 30% acetonitrile was quantified by BCA and visualized by means of SDS-PAGE. Immunoblotting was performed, using a subject's serum positive to this new allergen. The purified fractions correspond to an 18 kDa protein under non reducing conditions with a yield of 0.2 mg/g of PT pollen extract. Results of the SDS-PAGE and immunoblotting assay are shown in Fig 4.

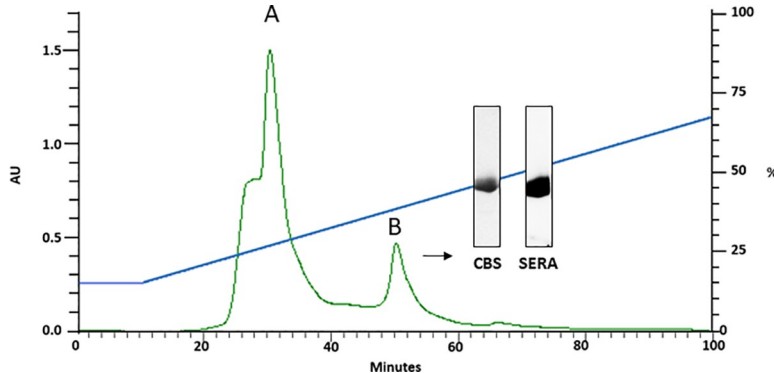

**Fig 4. HPLC purification profile of the 18kDa band Pru p 9 (peak B) and SDS-PAGE (CBS) and immunoblotting assay with a pool of sera.**

## Comparison of subject's SPT positive versus those negative to Pru p 9

Of the 62 cases sensitized to PT, 41% were SPT positive to Pru p 9 (Group A) and 59% were SPT negative to Pru p 9 (Group B). Results are shown in Table 1. Subjects from Group A were younger (29 versus 37 y.o.) (p<0.05) but no significant differences were found for gender. The number of pollens to which each subject was sensitized was similar in both groups. Most of the cases were atopic, with *Salsola kali* pollen being the mostly prevalent in Group A. Regarding clinical entities, the prevalence of conjunctivitis, rhinitis and asthma was higher in Group B, with differences for conjunctivitis and rhinitis (p<0.05 in both) but not for asthma.

## IgE detection of Pru p 9. Comparison between sera SPT PT+/Pru p 9+ and SPT PT +/Pru p 9-

Fig 5 shows results of sera from eleven cases who were SPT positive to PT pollen divided into two groups: also SPT positive to Pru p 9 (lanes 1 to 5) and negative to Pru p 9 (lanes 6 to 11). Bands of different molecular masses were recognized, but in the Pru p 9 negative cases (lanes 6

**Table 1. Comparison of subjects sensitized to pollen who were SPT positive (Group A) and negative (Group B) to Pru p 9.**

| Subjects (62) | Group A Pru p 9 + | Group B Pru p 9 - | Significance |
|---|---|---|---|
| Number | 26 | 36 | |
| Age | 29 (±13.2) | 37 (±12.7) | p <0.05 |
| Gender M/F | 61% | 61% | n.s. |
| Number of pollens | 3 (±2.06) | 3 (±1.7) | n.s. |
| *Phleum pratense* | 44.4% | 55.6% | n.s. |
| *Olea europaea* | 37% | 63% | n.s. |
| *Cupressus arizonica* | 44% | 56% | n.s. |
| *Salsola kali* | 45.2% | 54.8% | n.s. |
| *Platanus acerifolia* | 28.3% | 30.7% | n.s. |
| *Parietaria judaica* | 43.8% | 56.3 | n.s. |
| Conjunctivitis | 27.6% | 72.4% | p <0.05 |
| Rhinitis | 30.8% | 69.2% | p <0.05 |
| Asthma | 25% | **30%** | n.s. |

M: male. F: female. SPT: skin prick test. n.s.: not significant.

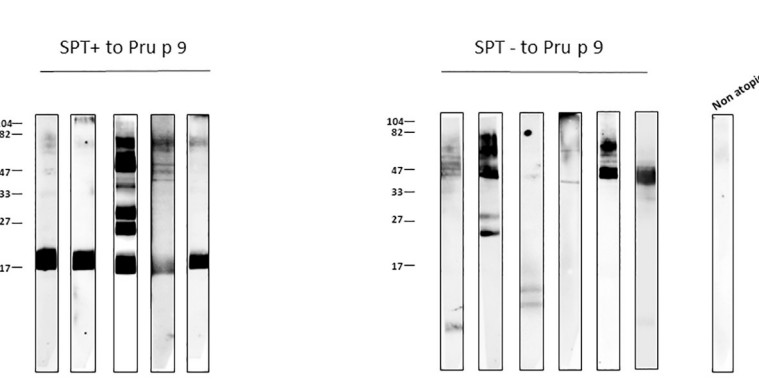

**Fig 5. Immunoblotting assay with sera of subjects with SPT positive to PT pollen.** From lane 1 to 5 they were also Pru p 9 SPT positive and from 6 to 11 Pru p 9 SPT negative.

to 11) the 18 kDa molecular band was not recognized by subject's IgE. No band was visible when a non-atopic serum was used as control.

## NPT with Pru p 9 in PT pollen sensitized subjects

To establish the involvement of Pru p 9 in the induction of symptoms, we performed NPT with PT pollen and Pru p 9 as a proof of concept. For this purpose, we challenged five cases with positive SPT to PT pollen and to Pru p 9 (Group A), five cases with positive SPT to PT pollen but negative to Pru p 9 (Group B) and the control group as detailed in the Materials and Methods section. The results presented as a decrease in the volume of the nasal cavity measured considering the basal value as 100% (Fig 6). In both groups there was a fall in the nasal cavity volume greater than 20% with PT pollen nasal provocation. In contrast, with Pru p 9 only the cases from Group A had a positive challenge. In all of them, typical symptoms were indicative of an immediate nasal response, although none of the subjects had symptoms corresponding to a dual or late response (Table 2).

The results of the control group are shown in S2 Fig. In two out of six cases a positive response was observed with a fall in the nasal volume of 25 and 30% respectively. No response was found with Pru p 9.

## Discussion

The data shown in this study provide important and new information about allergens of PT pollen, with specific focus on a novel allergen, named as Pru p 9 [20, 21] by the WHO-IUS

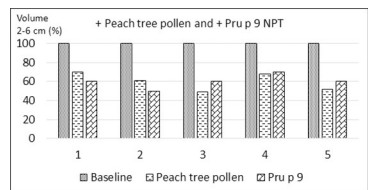
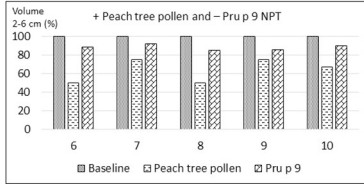

**Fig 6. Nasal provocation tests (NPT) with peach tree (PT) pollen and Pru p 9.** From 1 to 5 cases of Group A; from 6 to 10 cases of Group B.

**Table 2. Clinical characteristics of the cases included for NPTs.**

| C | Age | Gender | PT pollen SPT | Pru p 9 SPT | Symptoms |
|---|-----|--------|---------------|-------------|----------|
| 1 | 40 | F | 100 | 100 | Nasal obstruction, itching in the nose and eyes sometimes with wheezing from February to June with relapses in September-October. |
| 2 | 45 | F | 100 | 200 | Nasal obstruction with rhinorrhea from March to June often accompanied by wheezing. |
| 3 | 15 | F | 400 | 100 | Sneezing, rhinorrhea and nasal obstruction from February to June accompanied by itching in the eyes. |
| 4 | 62 | M | 100 | 110 | Nasal obstruction with itching and pruritus and rhinorrhea accompanied by pruritus and redness in the eyes through the year with more intensity from March to June. |
| 5 | 27 | M | 100 | 100 | Nasal obstruction with rhinorrhea and itching in nose and eyes when exposed to peach trees in flowering season. |
| 6 | 22 | M | 200 | - | Pruritus and erythema in the eyes with nasal pruritus and obstruction from February to early June. |
| 7 | 15 | F | 120 | - | Nasal obstruction with rhinorrhea and itching eyes accompanied by wheezing when visiting the farm in the spring. Control of bronchial symptoms with B2-agonist inhalers. |
| 8 | 45 | M | 200 | - | Itching and erythema in the eyes accompanied by nasal obstruction and sneezing during the whole year more intense from late June to late April. |
| 9 | 58 | F | 100 | - | Nasal symptoms in spring from March to June. |
| 10 | 44 | F | 190 | - | Nasal obstruction, rhinorrhea and sneezing accompanied by pruritus and eye redness in spring (May and June). |

NPT: nasal provocation test. C: case number. M: male, F: female. PT pollen SPT: skin prick test to Peach tree pollen. Pru p 9 SPT: skin prick test to Pru p 9. Negative result is shown as (-).

Positive SPT is shown as the percentage of the wheal area of histamine given the histamine value of 100.

nomenclature committee. It belongs to PR-1a protein family, relevant for the induction of respiratory and ocular symptoms in allergic patients, independently whether they were directly or indirectly exposed [28]. Although it has been shown that PT allergens presented in leaves and branches can induce both upper and lower airway symptoms in exposed workers, the association between symptoms and specific allergens from the pollen was not verified [14,15]. Although fruit trees' pollens including PT have been considered of low allergenic potency by their entomophilous character [2, 13], our data question these observations and are in line with other studies pointing to the same observation [10, 11].

In our study we used a well-defined model of PT pollen allergy with a large number of affected subjects as previously reported [12]. Most of our cases were atopic, with response to more than two pollens besides that of PT pollen. According to the clinical data, the period of the year when the affected subjects develop symptoms includes from late February to early May, months when PT is flowering and this pollination could overlap with those of grass and other weed pollens [29]. The two most common pollens involved in our cases were grasses (*Phleum pratense*) and olive tree (*Olea europaea*). Symptoms induced by the former may overlap with PT pollen but in this region olive tree pollen release occurs from the second to the third week of May, with concentrations lower than 400g/m$^3$ air [29, 30], suggesting that this does not contribute importantly to symptoms in this region.

Most cases in our study suffered from rhinitis, followed by conjunctivitis and in a lower proportion asthma, as reported for the classical allergic pollens [2]. Regarding asthma, our data differ from other studies that included subjects with allergy to ornamental flowers and fruits [5, 19]. In the study by Demir et al. dealing with allergy to rose pollen, the high frequency of asthma could be due to other factors, such as perennial allergens and close pollen exposure [5]. In our study, for all the subjects included, whether directly or indirectly exposed in agricultural and other activities, the data were similar in both groups, with no significant differences.

The occupational relevance of PT pollen has been studied previously and, with one exception where a relatively large group with occupational asthma was evaluated [15], the other

studies were limited to individual cases. One study showed a patient with occupational asthma to PT pollen, with the allergen involved being Pru p 3 that is present in leaves and branches [14] and a second one identified a 20 kDa (Glutathione S-transferase) allergen in a worker involved in artificial pollination [31].

Concerning the allergenic profile, it was notable that the 41% of the cases sensitized to PT pollen presented IgE antibodies to Pru p 9. This has been identified as a member of the Pathogenesis-related Protein family PR-1a, which is widely distributed in all plant species, but until now its allergenic capacity has not been reported in PT pollen. In the study carried out by Perez-Calderon *et al*, in subjects sensitized to PT leaves and branches, several proteins were shown to have allergenic activity, corresponding to bands of a molecular mass of 10 kDa, 14.5 kDa and others between 28–66 kDa, although no further identification was reported [15]. To our knowledge, detailed studies on fruit pollen have only been carried out with palm tree pollens [32, 33].

Regarding the other allergens identified, M2 was a glucan endo-1,3- beta glucosidase present in birch, ash and olive tree pollens, and M3 was a polygalacturonase also presented in grasses, olive tree, *Cupressus*, *Platanaceae*, *Salsola kali*, Japanese cedar, *Liliaceae* and other pollens that could contribute to the seasonal symptoms experienced by our patients. Regarding endo-1,3-beta glucosidase, its molecular mass differs from that of the intact enzymes (around 40 kDa). However, the presence of proteolytic fragments that have been assigned to allergens such as Ole e 4 has been detected for this allergen [34]. Regarding M4, UTP-glucose-1-phosphate uridyltransferase, we found no equivalent in the pollens to which our subjects were exposed.

This study was undertaken in a region known as the Ricote valley irrigated by the Segura River, with an estimated population of 250,000 inhabitants. Assuming a prevalence of 20% of sensitization in the general population [12] it is expected that a large number of people could be sensitized, with many of them having symptoms in the period of flowering [35,36]. The work covered not only the cultivar workers but it is indirectly extended to other exposed people including children. Furthermore, the village where this study was undertaken, Blanca, with 6,200 inhabitants, was the fifth in density of trees per hectare [37], with other villages having more trees per land hectare, which implies higher pollen exposure in the environment and more sensitization to PT pollen. In Spain there are other regions in the North-East, South and West with large extensions of PT cultivars and many people are therefore exposed to PT pollen. Accordingly, a larger population besides to exposed workers might have symptoms caused by this pollen. Regarding the worldwide population, the USA together with China, Italy and Spain, are the countries with the largest extension and production of PT crops. In the USA the highest producer states are California, South Carolina and Georgia, though PT are also cultivated in other areas [38]. In 2014 North Carolina led an international project to reveal the peach genome. This was an important scientific achievement that contributed to better understanding of plagues and infections and can be useful for the identification of new PT pollen allergens [39].

A weakness in this study could be a selection bias of the cases recruited for the study, however was unlikely because both allergic and non-allergic consecutive cases were included during the process of recruitment and the data were similar to the previous estimations [12]. Another is we assumed that PT pollen was the major inducer of allergy in this population because subjects developed symptoms during its flowering season and were exposed to grass and other weeds pollen. However, the purpose of our work was to identify PT pollen allergens focusing on Pru p 9 and to show the capacity of inducing symptoms by challenge a proof of concept.

Work in progress includes the quantitation of PT pollen in different areas close and distant to peach tree cultivars, an in-depth study of the molecular properties of Pru p 9 as well as protein characterization of the other allergens and the cross reactivity with other common environmental pollens and the clinical relevance in exposed population including the occupational implications.

## Conclusion

In summary, we provide for the first-time evidence supporting the role of PT pollen in inducing sensitization and allergy in an indirectly exposed population in an area where this fruit is highly cultivated. From this pollen, several allergens were recognized that may present cross-reactivity with allergens of other pollens prevalent in the area of study. One of the PT allergens, Pru p 9, was characterized and shown to elicit respiratory symptoms in the studied population.

## Supporting information

**S1 Fig. Mass spectrometry analysis of Pru p 9.** The identification was carried out in the proteomics service of Complutense University of Madrid. (https://www.ucm.es/gyp/proteomica).
(TIF)

**S2 Fig. Control group nasal provocation tests (NPT).**
(TIF)

**S1 Raw images.**
(PDF)

## Author Contributions

**Conceptualization:** Miguel Blanca.

**Data curation:** Miguel Blanca, Laura Victorio Puche, María Garrido-Arandia, Laura Martin-Pedraza, Alejandro Romero Sahagún, José Damian López-Sánchez, Carmen Galán, Antonio Marin, Mayte Villaba, Araceli Díaz-Perales, Maria Luisa Somoza.

**Formal analysis:** Miguel Blanca, Laura Victorio Puche, María Garrido-Arandia, Laura Martin-Pedraza, Alejandro Romero Sahagún, José Damian López-Sánchez, Carmen Galán, Antonio Marin, Araceli Díaz-Perales, Maria Luisa Somoza.

**Funding acquisition:** Miguel Blanca.

**Investigation:** Miguel Blanca, Laura Victorio Puche, María Garrido-Arandia, Laura Martin-Pedraza, Alejandro Romero Sahagún, José Damian López-Sánchez, Carmen Galán, Mayte Villaba, Araceli Díaz-Perales, Maria Luisa Somoza.

**Methodology:** Miguel Blanca, María Garrido-Arandia, Laura Martin-Pedraza, Alejandro Romero Sahagún, José Damian López-Sánchez, Carmen Galán, Antonio Marin, Mayte Villaba, Araceli Díaz-Perales.

**Project administration:** Miguel Blanca, Maria Luisa Somoza.

**Resources:** Miguel Blanca, Laura Victorio Puche, Laura Martin-Pedraza, Alejandro Romero Sahagún, José Damian López-Sánchez, Carmen Galán, Antonio Marin, Mayte Villaba, Araceli Díaz-Perales.

**Software:** Miguel Blanca, Laura Victorio Puche, Maria Luisa Somoza.

**Supervision:** Miguel Blanca, Laura Victorio Puche, María Garrido-Arandia, Alejandro Romero Sahagún, José Damian López-Sánchez, Carmen Galán, Antonio Marin, Mayte Villaba, Araceli Díaz-Perales.

**Validation:** Miguel Blanca, María Garrido-Arandia, Laura Martin-Pedraza, Alejandro Romero Sahagún, José Damian López-Sánchez, Carmen Galán, Antonio Marin, Mayte Villaba, Araceli Díaz-Perales, Maria Luisa Somoza.

**Visualization:** Miguel Blanca, Laura Martin-Pedraza, Carmen Galán, Mayte Villaba, Araceli Díaz-Perales.

**Writing – original draft:** Miguel Blanca, José Damian López-Sánchez, Carmen Galán, Mayte Villaba, Araceli Díaz-Perales, Maria Luisa Somoza.

**Writing – review & editing:** Miguel Blanca, José Damian López-Sánchez, Carmen Galán, Mayte Villaba, Araceli Díaz-Perales, Maria Luisa Somoza.

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
