## [Editor Report · Decision Letter 0]

7 Aug 2019

PONE-D-19-19788

Pru p 9, a new allergen eliciting respiratory symptoms in subjects sensitized to Peach tree pollen.

PLOS ONE

Dear Dr Maria Luisa Somoza,

Thank you for submitting your manuscript to PLOS ONE. After careful consideration, we feel that it has merit but does not fully meet PLOS ONE’s publication criteria as it currently stands. Therefore, we invite you to submit a revised version of the manuscript that addresses the points raised during the review process.

We would appreciate receiving your revised manuscript by October 6. To enhance the reproducibility of your results, we recommend that if applicable you deposit your laboratory protocols in protocols.io, where a protocol can be assigned its own identifier (DOI) such that it can be cited independently in the future. For instructions see: http://journals.plos.org/plosone/s/submission-guidelines#loc-laboratory-protocols

We look forward to receiving your revised manuscript.

Kind regards,

Davor Plavec

Academic Editor

PLOS ONE

2. Thank you for including your ethics statement: "The study was approved by our institutional Ethics Committee. All the participants have signed an informed consent."

3. Please provide additional details regarding participant consent. In the Methods section, please ensure that you have specified (1) whether consent was informed and (2) what type you obtained (for instance, written or verbal). If your study included minors, state whether you obtained consent from parents or guardians. If the need for consent was waived by the ethics committee, please include this information.

4. Thank you for your Financial disclosure statement "Supported by grant FIS PI17/00615, RIRAAF RD12/0013/0005 and ARADyAL RD16/0006/0024/ RD16/0006/0014."

Please expand the acronyms so that it states the name of your funders in full.

Additional Editor Comments (if provided):

The manuscript covers an interesting topic but prior to sending it for peer review it needs a major revision in the methods section. All the methods used and the recruitment proces for patients should be described in significantly more detail. Also there is a significant mistake in Table 2 regarding the significance of difference between 2 tested groups regarding clinical presentation (conjuctivitis, rhinitis, asthma).
---

## [Author Response · Author response to Decision Letter 0]

29 Oct 2019

Response: This has been changed accordingly in the “Revised manuscript”

2. Thank you for including your ethics statement: "The study was approved by our institutional Ethics Committee. All the participants have signed an informed consent."

Response: This has been amended.

Response: This has been made.

3. Please provide additional details regarding participant consent. In the Methods section, please ensure that you have specified (1) whether consent was informed and (2) what type you obtained (for instance, written or verbal). If your study included minors, state whether you obtained consent from parents or guardians. If the need for consent was waived by the ethics committee, please include this information.

Response: This has been added.

4. Thank you for your Financial disclosure statement "Supported by grant FIS PI17/00615, RIRAAF RD12/0013/0005 and ARADyAL RD16/0006/0024/ RD16/0006/0014."

Please expand the acronyms so that it states the name of your funders in full.

Response: This has been expanded and included in the cover letter.

Response: The original blot/gel image data have been added as Supporting Information and noted in the cover letter.

Additional Editor Comments (if provided):

The manuscript covers an interesting topic but prior to sending it for peer review it needs a major revision in the methods section. All the methods used and the recruitment proces for patients should be described in significantly more detail. Also there is a significant mistake in Table 2 regarding the significance of difference between 2 tested groups regarding clinical presentation (conjuctivitis, rhinitis, asthma).

Response: In the “Revised manuscript” the methods section has been described precisely: methods used and recruitment of patients.

 The mistake in Table 2 has been corrected.

Reviewers' comments:

Response: Our figure files have been uploaded to PACE.

All the changes in the text have been highlighted in bold.

---

## [Decision Letter · Decision Letter 1]

13 Dec 2019

PONE-D-19-19788R1

Pru p 9, a new allergen eliciting respiratory symptoms in subjects sensitized to peach tree pollen.

PLOS ONE

Dear Dr Maria Luisa Somosa,

Thank you for submitting your manuscript to PLOS ONE. After careful consideration, we feel that it has merit but does not fully meet PLOS ONE’s publication criteria as it currently stands. Therefore, we invite you to submit a revised version of the manuscript that addresses the points raised during the review process.

We would appreciate receiving your revised manuscript by Jan 27 2020 11:59PM. To enhance the reproducibility of your results, we recommend that if applicable you deposit your laboratory protocols in protocols.io, where a protocol can be assigned its own identifier (DOI) such that it can be cited independently in the future. For instructions see: http://journals.plos.org/plosone/s/submission-guidelines#loc-laboratory-protocols

We look forward to receiving your revised manuscript.

Kind regards,

Davor Plavec

Academic Editor

PLOS ONE

Additional Editor Comments (if provided):

Dear Authors,

please revise your manuscript according to the reviewers' comments.

Reviewers' comments:

Reviewer's Responses to Questions

**Comments to the Author**

1. If the authors have adequately addressed your comments raised in a previous round of review and you feel that this manuscript is now acceptable for publication, you may indicate that here to bypass the “Comments to the Author” section, enter your conflict of interest statement in the “Confidential to Editor” section, and submit your "Accept" recommendation.

Reviewer #1: (No Response)

Reviewer #2: (No Response)

2. Is the manuscript technically sound, and do the data support the conclusions?

Reviewer #1: Yes

Reviewer #2: Yes

3. Has the statistical analysis been performed appropriately and rigorously? 

Reviewer #1: Yes

Reviewer #2: Yes

4. Have the authors made all data underlying the findings in their manuscript fully available?

Reviewer #1: Yes

Reviewer #2: Yes

5. Is the manuscript presented in an intelligible fashion and written in standard English?

Reviewer #1: Yes

Reviewer #2: Yes

6. Review Comments to the Author

Reviewer #1: 1. Summary and overall impression

The manuscript represents a comprehensive analysis of the etiology of the allergic sensitization in a population directly or indirectly exposed to peach tree allergens in the regions where this pollen is substantially cultivated. Importantly, it brings identification of allergen(s) responsible for pollen sensitization and respiratory allergic symptoms. Although the study has been performed using the renewed methods and does not utilize a new technology, it adds a great value to the field of allergy sensitization and atopy.

The Discussion contains a concise summary and relevance of the results. The results are extensively elaborated in the context of similar publications and explain how the obtained results fit into the other research groups' findings. The last paragraph should be extended regarding the impact on the future work, i.e. steps and ideas for further research should be proposed.

The manuscript is well structured and the scope of the project is clearly presented. It enables a reader a complete and easy progression through the work. A coherent presentation of the results is consequently easy to grasp. The accompanying figures are simple and illustrative.

2. Evidence and examples:

Objective

The objective is nicely presented in a given theoretical background and argumented well enough to demonstrate its relevance and purpose. Given the size of the peach tree cultivated area and the prevalence of peach tree sensitization, the motivation is more than justified and the potential future application of the research is awaited.

Originality

The work uses established models of allergy study design and although it is not innovative, it builds up new knowledge to the topic.

Literature review

The literature review is nicely presented. The literature overview regarding the methodology of the study could be supported by a few more similar publications.

Proposed research

The proposed research is reasoned and well structured.

Methodology

A major drawback in the experimental design is the lack of the control population group. The drawbacks in the methodology and similar methodology-related issues should be transparently declared.

Minor revisions:

Paragraph 2.3: Peach Tree Pollen protein extract and Pru p 9 purification from peach pollen

• In the methodology should be clearly stated that the two chromatographic steps were used to obtain purified Pru p 9. First step - ion exchange chromatography using Waters AccellTM Plus QMA Sep-PakR cartridge should also include the size of the pores of the used cartridge. The second step - reversed-phase chromatography is described in detail and clearly.

Paragraph: Analytical procedures: SDS-PAGE analysis

• Tryptic digestion of bands excised from gel should include information regarding digestion mixture, concentration, period of incubation, temperature etc. or at least refer to a previous reference or established protocol. Although, in-gel trypsin digestion is a standard method it is relevant to state specific methodological details related to this work and type of sample.

• For MS analysis besides the type of used spectrometer for clarity the authors should add additional details related to the type of targeted plates, sample/matrix preparation (volume); type of laser with which is equipped mass spectrometer (wavelength, frequency). Need to add details regarding spectra recording: over which m/z range (linear mode?), at which accelerating voltage, average number of individual laser shots.

Optional additional figure: Mass determination spectra (figure of representative MALDI-ToF MS spectra to be added to the table with aminoacids residues verified by sequence analysis).

Reviewer #2: 1. More specific results and conclusions should be provided in the abstract

2. The criteria for patient selection and inclusion in the study need to be clarified, and indicate the number of respondents selected by which criteria

3. Provide a reference for the SPT, as well as a procedure for PT Pollen extracts

4. Clarify why respondents were done NPT

5. How do you explain that exposure to a particular allergenic component of PT is manifested by symptoms due to sensitization to that component and not because of cross-reactivity, because you did not provoke with the cross-reactive allergens

6. Indicate the most important weaknesses of your manuscript.

7. PLOS authors have the option to publish the peer review history of their article (what does this mean?). If published, this will include your full peer review and any attached files.

Reviewer #1: No

Reviewer #2: Yes: Mirjana Turkalj

---

## [Author Response · Author response to Decision Letter 1]

14 Jan 2020

Review Comments to the Author

Reviewer #1: 1. Summary and overall impression

The manuscript represents a comprehensive analysis of the etiology of the allergic sensitization in a population directly or indirectly exposed to peach tree allergens in the regions where this pollen is substantially cultivated. Importantly, it brings identification of allergen(s) responsible for pollen sensitization and respiratory allergic symptoms. Although the study has been performed using the renewed methods and does not utilize a new technology, it adds a great value to the field of allergy sensitization and atopy.

The Discussion contains a concise summary and relevance of the results. The results are extensively elaborated in the context of similar publications and explain how the obtained results fit into the other research groups' findings. The last paragraph should be extended regarding the impact on the future work, i.e. steps and ideas for further research should be proposed.

The manuscript is well structured and the scope of the project is clearly presented. It enables a reader a complete and easy progression through the work. A coherent presentation of the results is consequently easy to grasp. The accompanying figures are simple and illustrative.

Response from the Authors:

The last paragraph has been extended with future work ideas (line 402-405).

2. Evidence and examples:

Objective

The objective is nicely presented in a given theoretical background and argumented well enough to demonstrate its relevance and purpose. Given the size of the peach tree cultivated area and the prevalence of peach tree sensitization, the motivation is more than justified and the potential future application of the research is awaited.

Originality

The work uses established models of allergy study design and although it is not innovative, it builds up new knowledge to the topic.

Literature review

The literature review is nicely presented. The literature overview regarding the methodology of the study could be supported by a few more similar publications.

Proposed research

The proposed research is reasoned and well structured.

Response from the Authors:

According to the suggestion, new references were added (22-24).

Methodology

A major drawback in the experimental design is the lack of the control population group. The drawbacks in the methodology and similar methodology-related issues should be transparently declared.

Response from the Authors:

A control group is now included: patients from an area where there are no peach tree cultivars. We performed skin prick test and nasal provocation test with peach tree pollen and Pru p 9. The results are highlighted in the text (line 188-190 Materials and Methods section and line 305-307 and 313-315 Results section) and a new figure was added (supplementary S3).

As mentioned in the previous version of the Manuscript, we used sera from subjects allergic to pollens but not to Pru p 9 as Pool 2 included in the text (line 101-103 Materials and Methods section) and Figure 1. These subjects did not recognize Pru p 9.

Moreover, we also performed immunoblotting with a non-atopic serum (line 295-296 Results section) as shown in Figure 4, which did not recognize any specific or unspecific band.

Minor revisions:

Paragraph 2.3: Peach Tree Pollen protein extract and Pru p 9 purification from peach pollen

• In the methodology should be clearly stated that the two chromatographic steps were used to obtain purified Pru p 9. First step - ion exchange chromatography using Waters AccellTM Plus QMA Sep-PakR cartridge should also include the size of the pores of the used cartridge. The second step - reversed-phase chromatography is described in detail and clearly.

Paragraph: Analytical procedures: SDS-PAGE analysis.

Response from the Authors:

This has been clarified in the text (line 110-117).

• Tryptic digestion of bands excised from gel should include information regarding digestion mixture, concentration, period of incubation, temperature etc. or at least refer to a previous reference or established protocol. Although, in-gel trypsin digestion is a standard method it is relevant to state specific methodological details related to this work and type of sample.

• For MS analysis besides the type of used spectrometer for clarity the authors should add additional details related to the type of targeted plates, sample/matrix preparation (volume); type of laser with which is equipped mass spectrometer (wavelength, frequency). Need to add details regarding spectra recording: over which m/z range (linear mode?), at which accelerating voltage, average number of individual laser shots.

Optional additional figure: Mass determination spectra (figure of representative MALDI-ToF MS spectra to be added to the table with aminoacids residues verified by sequence analysis).

Response from the Authors:

We have included a new section in Material and methods: Mass spectrometry analysis (line 123-155). In this new section we have explained in detail all the MS procedure including digestion mixture, concentration, period of incubation, temperature as well as sample/matrix preparation (volume) among others.

We have also added a supplementary figure (S2) with the Mass determination spectra of the PR-1a protein (Pru p 9) as an example.

Reviewer #2: 

1. More specific results and conclusions should be provided in the abstract.

Response from the Authors: 

The abstract has been extended with more results, also conclusions were added (line 30-33 and line 39-42).

2. The criteria for patient selection and inclusion in the study need to be clarified, and indicate the number of respondents selected by which criteria

Response from the Authors: 

The criteria have been clarified and the number of the subjects has been added (line 72-77 Materials and Methods section, line 206-209 Results section).

3. Provide a reference for the SPT, as well as a procedure for PT Pollen extracts.

Response from the Authors: 

The references for the SPT were added (References number 22, 23), the procedure for PT pollen extract clarified and a reference added (24).

4. Clarify why respondents were done NPT

Response from the Authors: 

This is already explained in line 188-190. Nevertheless, the relevance of the reason why we did NPT with PT pollen and Pru p 9 is now better explained.

5. How do you explain that exposure to a particular allergenic component of PT is manifested by symptoms due to sensitization to that component and not because of cross-reactivity, because you did not provoke with the cross-reactive allergens

Response from the Authors: 

We have included a control group of patients where PT pollen cultivars do not exist and data showed that although challenge with PT pollen could be positive, the nasal provocation test with Pru p 9 was negative. This data is included as Supplementary Figure (S3) and included in the text (line 188-190 Materials and Methods section and line 313-315 Results section).

6. Indicate the most important weaknesses of your manuscript.

Response from the authors: 

This has been included in the last paragraph of the Discussion section (line 395-401).

---

## [Decision Letter · Decision Letter 2]

20 Feb 2020

Pru p 9, a new allergen eliciting respiratory symptoms in subjects sensitized to peach tree pollen.

PONE-D-19-19788R2

Dear Dr. Somoza,

We are pleased to inform you that your manuscript has been judged scientifically suitable for publication and will be formally accepted for publication once it complies with all outstanding technical requirements.

With kind regards,

Davor Plavec

Academic Editor

PLOS ONE

Additional Editor Comments (optional):

The manuscript is acceptable for publication in its current form.

Reviewers' comments:

Reviewer's Responses to Questions

**Comments to the Author**

1. If the authors have adequately addressed your comments raised in a previous round of review and you feel that this manuscript is now acceptable for publication, you may indicate that here to bypass the “Comments to the Author” section, enter your conflict of interest statement in the “Confidential to Editor” section, and submit your "Accept" recommendation.

Reviewer #1: All comments have been addressed

Reviewer #2: All comments have been addressed

2. Is the manuscript technically sound, and do the data support the conclusions?

Reviewer #1: Yes

Reviewer #2: Yes

3. Has the statistical analysis been performed appropriately and rigorously? 

Reviewer #1: Yes

Reviewer #2: Yes

4. Have the authors made all data underlying the findings in their manuscript fully available?

Reviewer #1: Yes

Reviewer #2: Yes

5. Is the manuscript presented in an intelligible fashion and written in standard English?

Reviewer #1: Yes

Reviewer #2: Yes

6. Review Comments to the Author

Reviewer #1: The authors have implemented the suggestions, especially regarding the methodology, and, thus, substantially improved their study. The study design should be kept in mind when planning future experiments.

Reviewer #2: The authors have adequately addressed my comments raised in a previous round of review and have adopted this manuscript acceptable for publication. All data underlying the findings described in the manuscript in a clear and correct way, and supports the conclusions.

7. PLOS authors have the option to publish the peer review history of their article (what does this mean?). If published, this will include your full peer review and any attached files.

Reviewer #1: No

Reviewer #2: No

---

## [Editor Report · Acceptance letter]

24 Feb 2020

PONE-D-19-19788R2 

Pru p 9, a new allergen eliciting respiratory symptoms in subjects sensitized to peach tree pollen. 

Dear Dr. Somoza:

I am pleased to inform you that your manuscript has been deemed suitable for publication in PLOS ONE. Congratulations! Your manuscript is now with our production department. 

With kind regards,

on behalf of

Dr. Davor Plavec 

Academic Editor

PLOS ONE